# Sensing Offshore Aquaculture Infrastructures for Data-Driven Dynamic Stress Analysis

Juan Carlos Sanz-González *, Amalia Jurado-Mc Allister *, Mercedes Navarro-Martínez, Rosa Martínez Álvarez-Castellanos, Ivan Felis-Enguix, Yassine Yazid, Yahya El-Mansouri, Fernando De Miquel-Moral, Hamid Errachdi and Ana Juan-Lición

Centro Tecnológico Naval y del Mar (CTN), Fuente Álamo, 30320 Murcia, Spain;
mercedesnavarro@ctnaval.com (M.N.-M.); rosamartinez@ctnaval.com (R.M.Á.-C.); ivanfelis@ctnaval.com (I.F.-E.);
yassineyazid@ctnaval.com (Y.Y.); yahyaelmansouri@ctnaval.com (Y.E.-M.);
fernandodemiguel@ctnaval.com (F.D.M.-M.); erhamid2@gmail.com (H.E.); anajuanlician@hotmail.com (A.J.-L.)
* Correspondence: jcarlossanz@ctnaval.com (J.C.S.-G.); amaliajurado@ctnaval.com (A.J.-M.A.)

**Abstract:** The presence of escaped fish in aquaculture facilities as a result of harsh meteorological conditions (more pressing in the face of climate change) requires a better understanding of this dynamic behaviour through vigilant monitoring and validated numerical models. In this context, data from strain and stress sensors as well as meteorological and current sensors installed at an aquaculture farm in the Region of Murcia (Spain) were collected, processed and analysed. Among them, the first results on the relationship between load and current sensors are presented. Due to the complexity of the time series, various analyses were conducted to examine their interrelation, encompassing the regression analysis of raw data and data segmented into different time intervals. Through this analysis, it was observed that employing distinct time windows better elucidated the data variability. Furthermore, an optimal data window of 240 data points was identified, demonstrating a significantly improved explanatory power, with the coefficient of determination ($R^2$) increasing by approximately 0.8 depending on the section. This paves the way for optimising the monitoring features that must be carried out to relate cause-and-effect variables in the behaviour of these offshore infrastructures.

**Keywords:** offshore aquaculture; escapes; adverse climatic events; load sensors; current meters; linear regression; window data method

**Key Contribution:** This study makes a significant advancement in the field of offshore aquaculture management by providing detailed insights into the dynamic stress experienced by aquaculture infrastructures under varying oceanographic conditions in order to prevent fish escapes.

## 1. Introduction

In 2020, the global aquaculture industry achieved a historic milestone, reaching a record production of 122.6 million tons, valued at USD 281.5 billion [1]. Furthermore, projections anticipate a substantial upswing in marine animal production, consumption and trade by 2030, with an estimated surge to 202 million tons. This surge is attributed to the sustained growth and expansion of the aquaculture sector [1]. As a result, the transformative trajectory of the marine aquaculture sector entails a shift towards establishing production facilities in the open sea [2]. In these offshore locations, the oceanographic and meteorological conditions pose heightened challenges [3]. The vulnerability of key structural elements, particularly mooring lines and nets, is significantly elevated compared to installations situated in more sheltered areas near the shore [4].

In numerical terms, just in Norway between 2010 and 2018, a total of 305 escape incidents were documented, involving Atlantic salmon (*Salmo salar*) and rainbow trout (*Oncorhynchus mykiss*), accounting for 1,960,000 registered escapes [5]. The Mediterranean

region faced a more substantial challenge, with 7,645,700 fish escaping from sea cages between 2007 and 2009, while the UK reported 1350 escapes during the same timeframe. Moreover, the associated financial toll (based on the value at the point of first sale) of these escapes in Europe was estimated at approximately EUR 47.5 million annually [6].

### 1.1. Effects of Escapes

It is imperative to recognise that escapes not only entail significant economic repercussions for aquaculture practitioners but also pose severe ecological, genetic, pathogenic and socioeconomic consequences, as underscored [7]. For example, escape incidents heighten the potential for disease and parasite transfer and contribute to their amplification within aquaculture settings. Indeed, it was demonstrated [8] that escapes from salmon aquaculture in Norway in 2021 served as reservoirs for sea lice in coastal waters. Public health concerns may arise post escape, as farmed fish are often medicated, potentially resulting in escaped fish containing active substances [9]. Additionally, escapes can exert profound effects on wild populations. The hybridisation of farmed fish with their wild counterparts has the capacity to genetically alter populations, diminishing local adaptation and adversely impacting population viability and integrity [10]. Cultivated fish, accustomed to consuming feed pellets in sea cages, undergo a dietary shift to natural prey upon escape. This adaptation introduces the potential for competition with local counterparts or other wild species for food and habitat [11]. For instance, escaped seabreams have been captured in fishing grounds and habitats, such as seagrass, sand, or rocky bottoms, where their wild conspecifics reside, thus preying on natural species [12]. Moreover, the impacts are particularly challenging if the escaped species are not native from the area [13], as they can lead to alterations in habitat complexity [14].

The aftermath of escape incidents extends to conflicts between the aquaculture and fishery industries. Escaped species are frequently captured by fisheries, resulting in heightened tensions and increased catches in local fisheries [12]. The comprehensive understanding and management of escape incidents are crucial for mitigating their ecological, economic and public health ramifications.

### 1.2. Causes of Escapes

Regarding the causes of escapes, the documented incidents in recent years include the wrecking of installations, inadequate technical conditions of the facilities, human error, predation by predators, collisions, poor inspection and working procedures, the lack of control systems and a deficit in competence among salmon farmers, as outlined in the strategy for sustainable aquaculture [6,7]. Among these causes, storms emerge as the most significant catalysts for structural failures, leading to low performance in moorings and cages and subsequent escape incidents as well as net holes [5].

Moreover, it is expected that climate change will further exacerbate the threat to offshore aquaculture [15]. Storms are expected to escalate in frequency and severity due to climate change, posing a substantial risk to aquaculture operations [6,7]. For instance, the storm Gloria in January 2020 along the Mediterranean coast surpassed previous events with unprecedented wave heights, durations and storm intensities, challenging established understanding of the wave climate in the Spanish Mediterranean [16]. This event, in turn, led to substantial damage to fish farms, freeing millions of fish into the wild [17].

### 1.3. Mitigation Measures

Among mitigation measures to avoid fish escapes, the implementation of a rigorous monitoring plan for all components, including anchorages, frameworks, buoys and both deep and surface anchors, has become a common practice in the fish farms. Nonetheless, these practices should be executed with sufficient frequency to detect potential issues and enable timely repairs or replacements before system failure, which is expensive and time-consuming [18]. In contrast, weather prediction models and dynamic simulations have been developed [19–21]. However, these models are not exempt from criticism. Some

of them do not adjust exactly to reality, and empirical experiments in real scenarios must be performed to demonstrate the models [22].

To prove the developed model, in situ experiments should be performed, but they are expensive and difficult to carry out. For instance, in Norway, full-scale commercial sea cage experiments assessing net deformation revealed that currents have the potential to diminish the net volume by 20–40%, underscoring the pivotal role of currents as a significant factor influencing the behaviour of sea cages [23]. Likewise, an empirical study also located in a Scandinavian county revealed that the presence of the sea cage significantly reduces the flow near the area and increases the turbulence in the upper water column [24].

The scarcity of empirical studies on sea cages, with most of them being concentrated in Norway, and the absence of comprehensive data for comparing hydrodynamics models leave many studies incomplete and uncertain. Moreover, the paucity of experiments in the Mediterranean exacerbates the lack of knowledge, impeding precision and security in the installation of the offshore sea cages, and therefore elevating the risk of structural failures in these areas where heavy storms are becoming more and more frequent [16].

Thus, this paper goes a step further beyond the advancements and developments previously made at CTN-Marine Technology Center [25,26] introducing a technology whose objective is to decrease the risk of breakages in aquaculture sea cages. By deploying a cutting-edge method to assess the infrastructure's condition and quantify potential breakages, this innovative approach marks a significant advancement in enhancing the safety and reliability of aquaculture installations.

## 2. Materials and Methods

### 2.1. Study Area

The pilot was developed in one of the meagre farms located in the Region of Murcia (Spain). The sea cage chosen was located 9 km from the coast and 45 m deep with strong exposure to waves and currents. This sea cage contained meagre, a species that, with its movement, can cause deformations in the net due to its large size, which can reach 2 m in length [27]. The net pen also contained an anti-current ring to prevent excessive movement due to current.

### 2.2. Sources and Data Collection

For monitoring the sea-cage infrastructure, four load cells in each of the main mooring lines and six net-moving sensors were deployed in the net-pen. In addition, oceanographic parameters were monitored with an oceanographic buoy, and data from the marine currents were collected using a current meter. In Figure 1, these sensors and their arrangement can be seen on the network.

In the present paper, we focus on the results obtained by two of the load cells (namely load sensors 1 and 3), and the data obtained in the current meter. Load cells 2 and 4 stopped working immediately after the installation and, therefore, the data could not be used.

The load cells were installed in the joints between the mooring lines and the mooring buoys. Load cell number 1 was deployed facing northeast, continuing with 2, 3 and 4 in a clockwise direction in a symmetric order with a 28.29 m sea cage diameter. The data were transmitted through a cable to the data logger installed on the surface, recorded on an SD card and removed during the field work. The load cells have a maximum weight that can support of 85 t.

The current meter was deployed to 27.5 m depth under the sea cage and facing up. The current meter collected data on current speed [m/s] and direction [°] every 2 m in the water column. The data were stored in internal memory and collected each time we went to the field.

Data collection period spanned during summer from 23 June to 8 August 2023. The sampling rate varied depending on the sensor. Load cells transmitted data to a data logger every second, while the current meters recorded data at 20 min intervals.

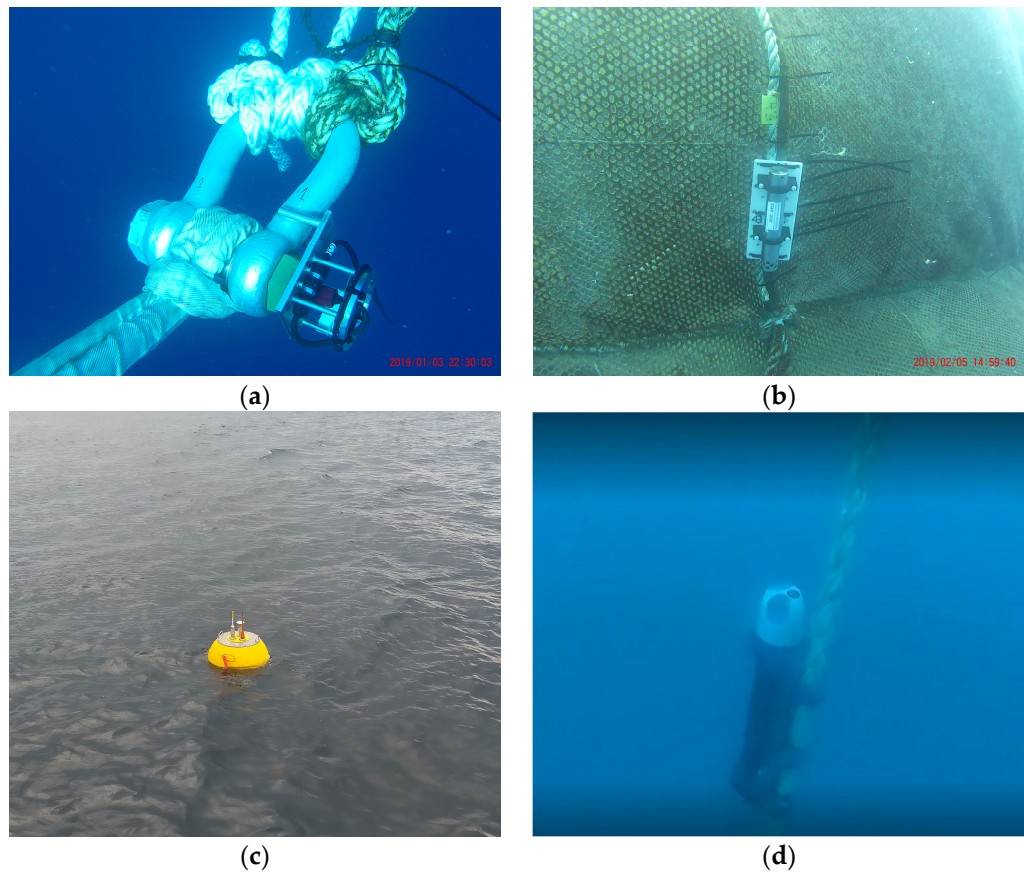

**Figure 1.** In situ monitoring devices within a fish farm cage located in the Region of Murcia, Spain. (**a**) Load sensor, (**b**) network sensor, (**c**) buoy and (**d**) current meter.

*2.3. Statistical Analysis*

In this study, data were collected from various sources, encompassing oceanographic variables, as well as sensor data from a marine aquaculture facility. The data types included load measurements (from load cells) and current data (from different current meters).

2.3.1. Preprocessing

Initially, we performed data cleaning to rectify inconsistencies and remove anomalies. This included the identification and imputation of missing values, ensuring no data point was disregarded without due justification. To address the issue of outliers and reduce noise, we employed an Interquartile Range (IQR) method for the load sensor features. Then, observations outside ±1.5 IQR bounds were deemed non-representative of the underlying pattern and thus excluded from further analysis. A proportion of 1% of data were removed using this method.

Special attention was given to synchronising the load and current meter data to ensure temporal alignment each minute. However, all datasets were aggregated by the hourly mean, facilitating analysis and reducing data volume.

2.3.2. Processing

Our data processing methodology was compartmentalised into three distinct sections, each addressing a separate aspect of this study. The sections were constructed to elucidate the interplay between sensor load data and current velocities, with the ultimate aim of enhancing our understanding of the underlying physical phenomena.

- Sensor comparative analysis. In the first section, we conducted a temporal analysis for each sensor individually. Time series graphs were generated to delineate the behaviour of each sensor over the study period. This visual inspection facilitated the identifi-

cation of any anomalous or periodic behaviour that warranted further investigation. Subsequently, a comparative analysis was initiated wherein the load data from two opposing sensors were juxtaposed in a scatter plot. A quadratic regression model was fitted to these paired data to capture any non-linear relationship between the sensors' readings. This model was selected based on preliminary analyses that suggested a quadratic relationship offered the best fit, thereby enabling the characterisation of the load response with greater fidelity than a linear model.

- Descriptive analysis of currents. The second section was dedicated to a descriptive analysis of the current velocities and their association with sensor load data. A correlation matrix was constructed encompassing all current velocities variables, allowing us to quantify the degree of linear relationship between current velocities at different depths. The matrix was extended to include the sensor load data, aiming to reveal any potential correlation between the dynamic behaviour of the currents and the sensor loads. This comprehensive analysis served to identify patterns and relationships that might not be readily apparent from isolated data points.

- Windowed data analysis for regression enhancement. The final section of our data processing involved the application of a windowing technique to the dataset. Data windows were established with the intention of refining the accuracy of the regression models. By segmenting the data into smaller subsets based on time intervals (data per minute), we aimed to enhance the granularity of our analysis. This approach allowed us to investigate whether the inclusion of more localised data subsets could explain a greater variability in the sensor data in relation to the currents. The size and overlap of the windows were methodically determined to balance the model's sensitivity to temporal variations against the risk of overfitting.

### 2.3.3. Postprocessing

The residual plots were meticulously scrutinised to detect any patterns or systematic deviations that could indicate violations of the model assumptions. The analysis of residuals also extended to verifying homoscedasticity and ensuring that the residuals were randomly distributed and independent of the predicted values for most models, which is paramount for the reliability of the regression models.

Upon completion of the regression analyses, we proceeded to a detailed postprocessing phase to thoroughly examine the significance of the regression coefficients and to evaluate the adequacy of the model fits. Post hoc analyses were conducted to assess the statistical significance of the regression coefficients. This involved computing *p*-values using the standard t-tests for each coefficient at the level of significance of 0.05.

## 3. Results

In this work, we have undertaken a comprehensive analysis of load data from two distinct sensors. Initially, we collected an extensive dataset comprising over 2.5 million individual readings across 16 variables which include the loads from both sensors and 13 current velocities from the surface to 27.5 m. To enhance the clarity and interpretability of the trends, we have aggregated these data points into hourly averages, thereby condensing the dataset to a more manageable 716 data points for each sensor. This reduction allowed for a more streamlined and focused examination of load variations over time.

### 3.1. Sensor Comparative Analysis

This scatter plot tracks the load data captured by the first sensor over time (Figure 2). The vertical axis represents the load and the horizontal axis denotes the date in month and year (MM-YY) format. Due to some issues with the weight sensors calibration, it has been decided to express the load in arbitrary units [a.u.]. The data points exhibit a positive trend, suggesting a gradual increase in load over the examined time frame. Despite some variability, the general direction is upwards, with the load increasing from below 1000 [a.u.] to nearly 3000 [a.u.].

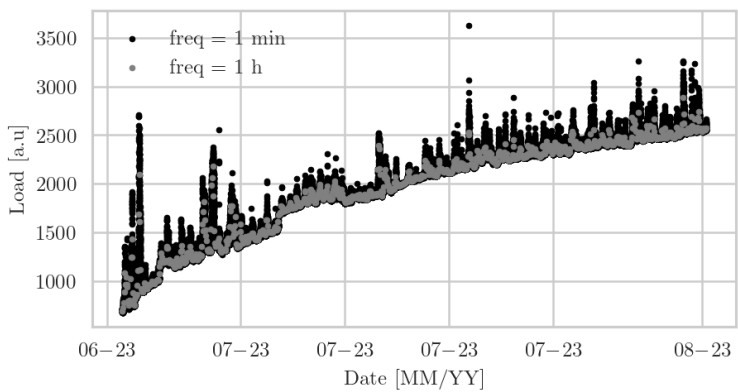

**Figure 2.** Load sensor 1 represented every minute (black) and every hour (grey).

The second scatter plot illustrates the readings from the third load sensor across the same temporal span (Figure 3). The trend in this graph is downwards, with the load decreasing from around −250 [a.u.] to below −1500 [a.u.]. The data points are more clustered in the early part of the graph and spread out as the load decreases.

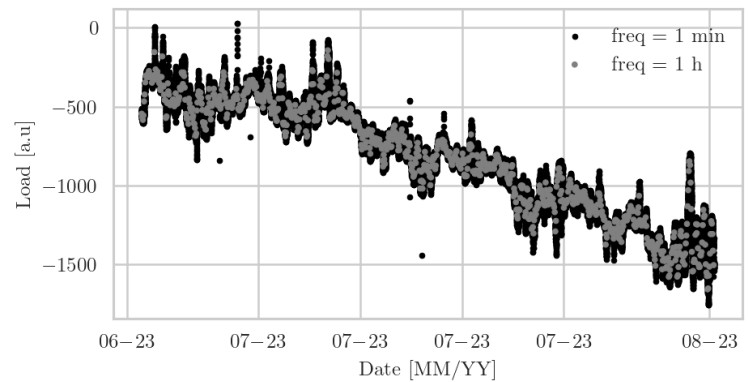

**Figure 3.** Load sensor 3 represented every minute (black) and every hour (grey).

Figure 4 presents a scatter plot that compares the loads recorded by load sensor 1 (*x*-axis) and sensor 3 (*y*-axis). It can be shown that below 1500 [a.u.] of sensor 1, the response of sensor 3 is nearly flat but, for higher loads, there is a quadratic relationship between them. Indeed, the data from both sensors are correlated (−0.86). They also represent a black curve with an adjusted second-degree polynomial.

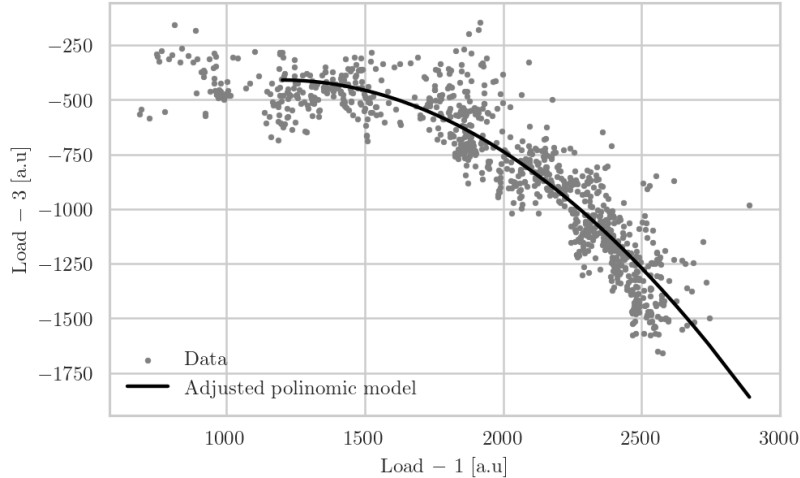

**Figure 4.** Adjusted quadratic relationship between both sensors.

The quadratic regression model applied to the relationship between the loads detected by the two sensors is characterised by the following equation:

$$y = ax^2 + bx + c \tag{1}$$

The coefficients obtained from the model fitting are as follows, $a = 1.21$, $b = -5.05^{-4}$ and $c = -1131.39$, where $x$ represents the load from sensor 1, and $y$ corresponds to the load from sensor 3. The model's intercept at $-1131.39$ sets the initial calibration for the relationship. The mean squared error (MSE) of the model is 18,733.29, which quantifies the average squared difference between the observed actual outcomes and the outcomes predicted by the model. Nonetheless, the coefficient of determination, or $R^2$, is 0.8446, suggesting that approximately 84.46% of the variance in the sensor 3 load can be explained by the quadratic model based on sensor 1's load. This strong $R^2$ value indicates a high level of predictive power and a substantial correlation between the two sensors' load measurements, validating the model's effectiveness in capturing the underlying relationship within the data.

### 3.2. Descriptive Analysis of Current Velocities

The initial analysis revealed an absence of direct correlation between the data obtained from load sensors and the measurements of water current at varying depths, except for the contiguous currents which show a maximum correlation of 0.63.

The correlation matrix heatmap (Figure 5) delineates the degree of linear relationship between sensor loads (Load 1—NE, Load 3—SW) and current velocities at varying depths (velocities at surface to velocities at 24.5 m). Notably, the contiguous current speed readings exhibit relatively high positive correlations, suggesting a pattern of coherent movement among adjacent water strata. This coherence likely reflects the influence of uniform hydro-dynamic forces acting upon proximate depths. In contrast, the sensor loads (Load 1—NE, Load 3—SW) manifest a pronounced negative correlation, indicating an inverse relationship between the loading conditions detected by the two sensors. Such a negative correlation could be indicative of differing load responses to the same environmental stimuli or may reflect the positioning and orientation of the sensors relative to the current flow.

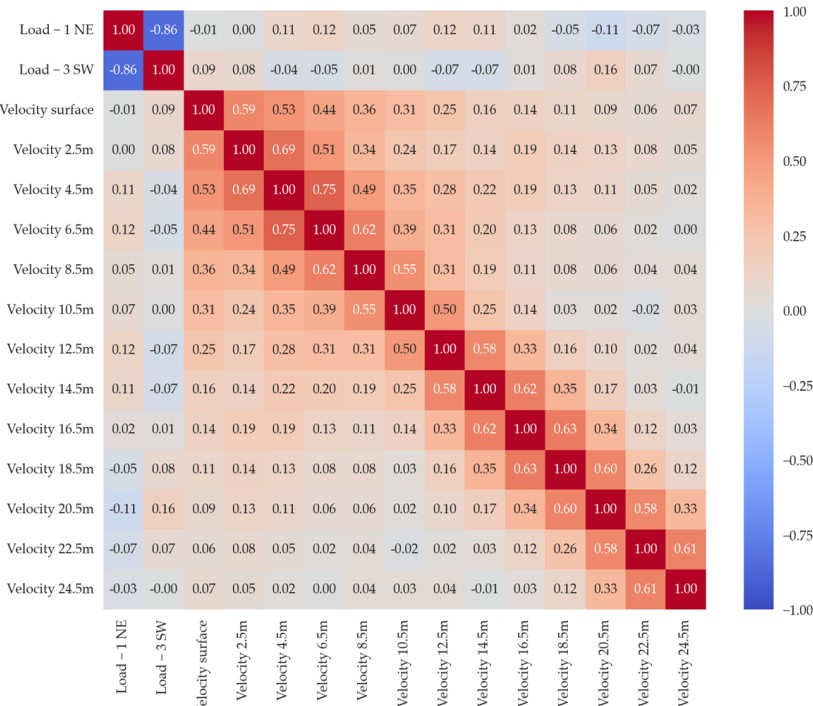

**Figure 5.** Correlation matrix of load sensors and velocities.

In the observed dataset, a notable discontinuity is present at a depth of 12.5 m. The collected data at this juncture indicate a distinct shift in the current speed, a phenomenon that coincides with the location of the thermocline. The thermocline is characterised by a rapid change in water temperature with depth, which can significantly affect the water's density and, consequently, its movement. The alignment of this sensor's data with the expected position of the thermocline suggests that the thermocline's presence at this specific depth influences the current speed. This confluence is critical for understanding the stratification of water columns and the dynamic interactions between temperature and current flow.

### 3.3. Load Sensors with Current (Windows)

To test the relationship with other metrics, a univariate linear regression model was run between the sensors and each of the velocities, giving a maximum $R^2$ of 0.008. A multivariate linear regression model was then run between sensor 1 and all currents and sensor 3 and all currents, giving a $R^2$ of 0.043 and 0.041 respectively.

These indicate a non-relationship between the load supported by the aquaculture net and the currents at different depths. To delve deeper into the relationship between these datasets, a multivariate regression analysis was performed by taking different time intervals (60, 120, 180, 240 and 300). The results indicated an increase in the coefficient of determination ($R^2$) as the time interval increased, reaching values of 0.8. However, a slight decrease in the $R^2$ coefficient was observed when employing a 300-unit time interval. Nevertheless, the $R^2$ coefficient remained higher than that observed in the original dataset.

This minimal difference can best be seen by looking at the distribution of $R^2$ for each of the sensors for a window of 240, as shown in Figure 6. Here, it can be seen that the calculated deviations around the data have a higher value, which is because most of the $R^2$ values are higher.

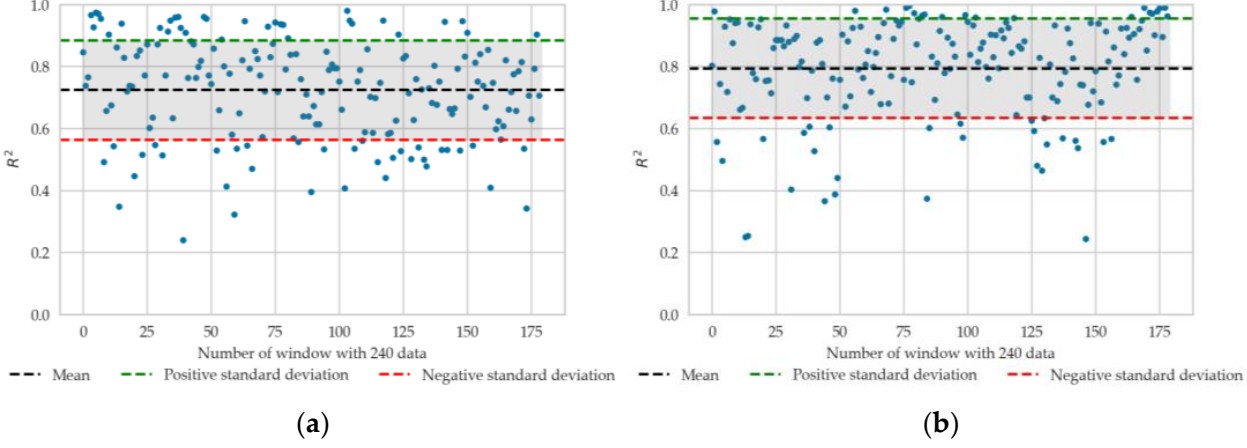

(**a**)                                      (**b**)

**Figure 6.** (**a**) Distribution of $R^2$ with 240 window data points, load sensor 1; (**b**) distribution of $R^2$ with 240 window data points, load sensor 3.

The optimal window discussed can be seen in the following figure, where it can be observed that as the size of the window increases, the distribution of $R^2$ becomes larger. This holds true up to the optimal window (240), after which it slightly decreases at 300. The fact that this is the case for both load sensors, i.e., that both have the same trend, is due to their high inverse correlation.

To check this in a more interpretable way, the distribution of the window residuals of the 60, 120, 240 and 300 data points is analysed. In the distribution of $R^2$ (load sensor 1) in Figure 7, it was observed that for window 60, most of the values are around 0.5 and increase as we increase the window, with the majority of $R^2$ being greater than 0.6 for window 240.

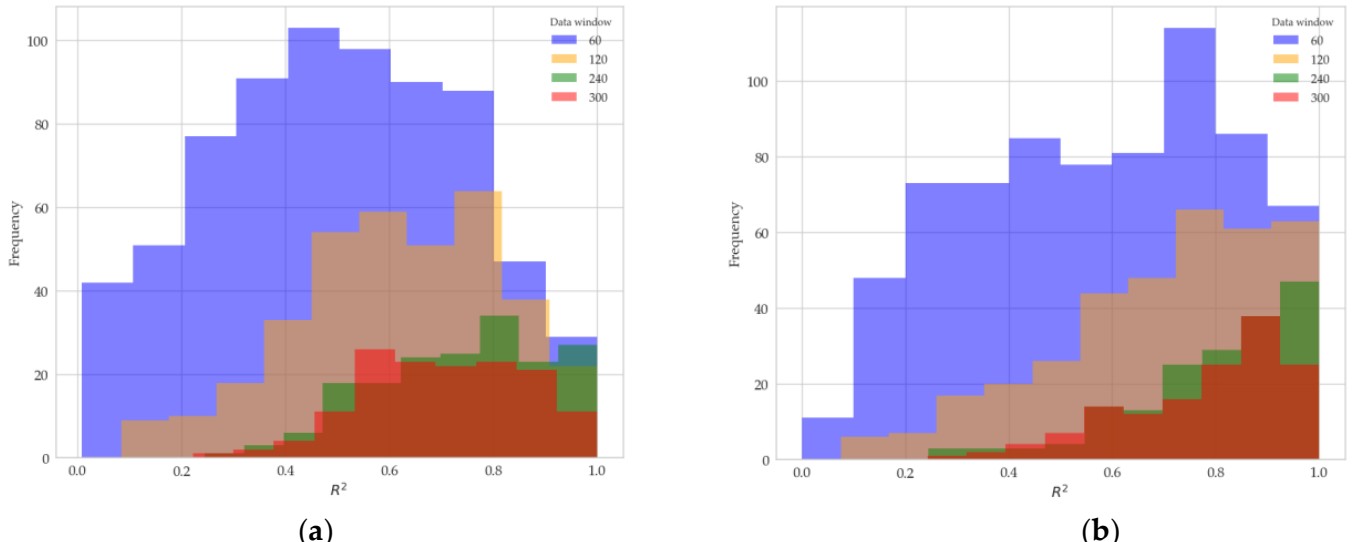

**(a)**                                                                                    **(b)**

**Figure 7.** (**a**) Distribution of $R^2$ with 60, 120, 240 and 300 interval data points, load sensor 1; (**b**) distribution of $R^2$ with 60, 120, 240 and 300 interval data points, load sensor 3.

In Figure 7b, the behaviour of load sensor 3 shows the same trend as already mentioned for load sensor 1. A small difference can be seen in the $R^2$ values, which are slightly higher for sensor 3, indicating that it explains the variability better, although the difference is minimal.

After extensive analysis, it was determined that the optimal windows to capture the variability of the data were 240 (Figure 8). These intervals showed a significant increase in the metric $R^2$ compared to the original dataset, meaning that the relationship between load sensors and water flows was better captured. Showing that there is no evidence of correlation in the previous analysis, multiple regressions are proposed to find better data uses.

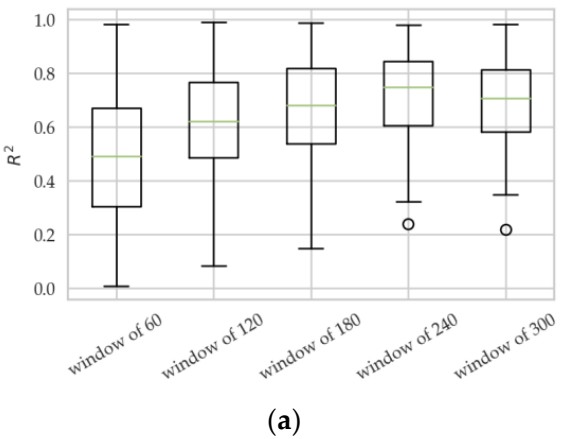 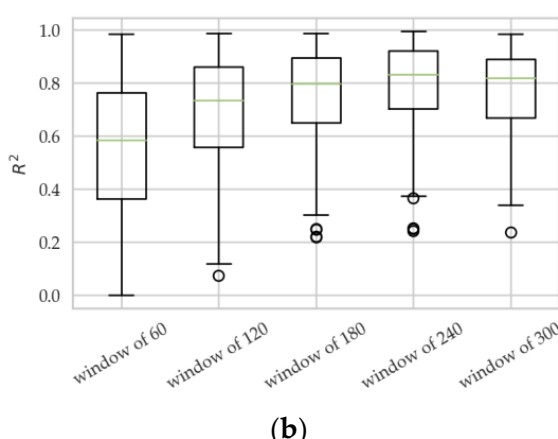

**(a)**                                                                                    **(b)**

**Figure 8.** (**a**) Boxplot load sensor 1; (**b**) boxplot load sensor 3.

## 4. Discussion

Load sensors 1 and 3 exhibit a strong negative correlation. This negative correlation might suggest that when one sensor is experiencing a high load, the other tends to experience a lower load, or vice versa. This could be due to the sensors being positioned on different parts of a structure that experiences differential loading depending on the current speed and direction. It is also crucial to consider the magnitude of the correlations, as very small positive or negative values (close to 0) may not represent a meaningful relationship in a practical sense. Moreover, other factors such as the time of day, seasonal variations,

geographical location and other geophysical variables could affect the currents and sensor readings [28], which should be accounted for in a comprehensive analysis.

Load sensor 1 has supplied normalised data, exhibiting an initial deviation of $\pm 85$ kg (0.83 kN). The recorded stress fluctuated between 593 kg (5.81 kN) and 809 kg (7.85 kN), showcasing an ascending trend throughout the sampling period. Conversely positioned on the opposite side, load sensor 3 affirms the data's normality by capturing negative values akin to the positive readings of load cell 1, reaching a maximum support of 224 kg (2.2 kN). This supports the accurate initial interpretation of CTN opposition, i.e., negative tension in the one not subjected to force and positive tension in the opposite, which bears some kind of load. To confirm this possible cause, this observed behaviour of opposing sensors will need to be compared with repeated situations under similar conditions.

The load sensors are supported in the following way: at one end, we have the anchoring line formed by PROFLEX [29] rope with eight strings with a total Ø of 62 mm and breaking load of 665 kN, from which the grid lines (reticular grid of 50 m × 50 m) are formed by PROFLEX strings of eight ropes with a total Ø of 72 mm and breaking load of 885 kN (at the vertices of one of the grids is where the load sensors have been installed). At the other end of the load sensor are the cage mooring ropes, made up of a total of twelve ropes, three in each corner and thus three at the other end of each load sensor. The material of each of the twelve ropes is 8-string Nylon with a total Ø of 38 mm and breaking load of 269 kN.

The tension forces recorded by these two opposing sensors, 1 and 3, seem to indicate that during the sampling period, the fastenings and structures were not subjected to risky situations, as they did not exceed 10 kN. This is significantly below the breaking load data for the cables used, which are 269 kN, 665 kN and 885 kN, as provided by the manufacturer. Nevertheless, the experiment was performed during the summer season; therefore, more measurements during at least one year would give better insight into the efforts supported by the sea cage.

The importance of testing the drag forces of a full-scale net cage lies in the fact that they usually do not adjust to the models. Fredriksson and colleagues showed that when strong currents exist, the drag on the cage increases tension in the mooring lines restricting the cage's horizontal motion and influencing the nonlinear component of the surge response. As a result, the real behaviour was not presented in the numerical models [22]. Moreover, the same study showed that the models used were usually more conservative and the load cell values were higher compared to field observations, probably due to the increase in viscous effects associated with lower Reynolds numbers at the model scale. Additionally, another study analysing the drag forces realised that values derived from full-scale net cage testing, when converted from model-scale testing, surpassed those estimated based on depth data. Nevertheless, conversely, the converted cross-sectional areas from model-scale testing were found to be smaller than the estimated values obtained in full-scale testing [30], showing again that numerical models and empirical studies do not always adjust to reality. If, as shown in the data, sensor 1 tends to support more load, a greater reinforcement of the ropes that are attached to the bottom line of the northeast face would be beneficial. In addition, costs can be saved on the southwest face, which does not receive as much impact. Information of this nature is crucial to work towards a precision aquaculture which aims to optimise resources and enhance the overall efficiency in the cultivation process, thereby promoting sustainable practices and mitigating environmental impacts.

The correlation between the adjacent currents is slightly high, but as they move away, the correlation decreases noticeably. This shows that depths that are close to each other have similar velocities, while as soon as the distance increases, the velocity between the currents becomes distinguishable. It has been shown that in the same area, different depths have different values [31], considerably affecting the force supported by the components of the sea cage at different parts (top vs. bottom). This again emphasises the necessity to fully determine the oceanographic parameters at different depths in a specific area using real data to obtain more precise models.

## 5. Conclusions

The results indicate that the relationship between the load on the underwater network and the current velocities is complex and influenced by the temporal scale of measurement. However, the negative correlation between the two sensors suggests a significant inverse relationship, indicating dynamic interactions within the underwater monitoring system. The spatial correlation between contiguous currents declines with distance, highlighting variable current conditions across depths, particularly around the thermocline. The lack of correlation between sensors and current velocities implies that other factors may influence the infrastructure's load. Regression analysis with current velocities as predictors yielded low $R^2$ values; however, the window data method seems to be the best method to explain these results and could be used in the future for further analysis, adding new oceanographic variables. This paper emphasises the significance of conducting experiments in authentic, real-world scenarios to comprehensively grasp the dynamics of offshore aquaculture sea cages. This approach is essential to prevent fish escapes and the associated adverse outcomes highlighted in the introduction. Our proposal suggests the preliminary installation and monitoring of a single sea cage using load cells before establishing a complete offshore facility. This step allows an assessment of whether the specific area meets the necessary parameters, ensuring a secure installation aligned with the financial resources of the aquaculture company.

**Author Contributions:** Methodology, A.J.-M.A.; formal analysis, J.C.S.-G., M.N.-M., Y.Y., Y.E.-M., F.D.M.-M., H.E. and A.J.-L.; data curation, J.C.S.-G. and M.N.-M.; writing—original draft, J.C.S.-G., A.J.-M.A., M.N.-M. and R.M.Á.-C.; supervision, I.F.-E. All authors have read and agreed to the published version of the manuscript.

**Funding:** This research was supported by the DIGI SAFE CAGE project, which is funded under the grants for consortia of entities engaged in investment and reform projects in research for technological development, innovation and the supply chain balance in the fishing and aquaculture sector. This initiative is part of the Recovery, Transformation and Resilience Plan (RTRP) as stipulated in the Royal Decree 685/2021 of 3 August from the Spanish Government and NextGenerationEU fundings.

**Institutional Review Board Statement:** This research did not involve human participants or animals. As such, ethical approval from an Institutional Review Board (IRB) or Ethics Committee was not required. The study exclusively utilized data obtained from the load cells installed in the sea cage's mooring lines, and current velocity acquired from the current meter deployed in the area.

**Data Availability Statement:** The data presented in this study are not publicly available, due to privacy reasons.

**Acknowledgments:** The authors would like to express their gratitude for the support provided by the partners of the DIGI SAFE CAGE project for their valuable contributions. Special thanks are also extended to the aquaculture company that made their facilities available for conducting the pilot study.

**Conflicts of Interest:** The authors declare no conflicts of interest.

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
