# Peer review of "Sensing Offshore Aquaculture Infrastructures for Data-Driven Dynamic Stress Analysis"

_fishes, doi:10.3390/fishes9020061_

Round 1

Reviewer 1 Report

Comments and Suggestions for Authors

The manuscript does provide important information to approach a delicate environmental problem that affect producers in Europe. The manuscript is well written and the matherials and methods are adecuate, results and conclusions suggest your technology may improve economic performance of marine farms by potentially prevent escape of the fishes. Minor corrections are needed to continue the publication process:

1. The numeration of the subsections in the introduction are wrong, please correct.

2. There is a phrase in chinese that indicates an error, please correct the error.

3. In the discussion the first three paragraph does not have any citation, at the last sentence of each of these paragraph you write some interesting conclusions to explain previous information, if possible include references to support your conclusions.

Please use the attached file to the minor corrections

Author Response

For research article

Response to Reviewer 1 Comments

1. Summary

Thank you very much for taking the time to review this manuscript. Please find the detailed responses below and the corresponding corrections highlighted in the re-submitted files.

2. Questions for General Evaluation

Reviewer’s Evaluation

Response and Revisions

Does the introduction provide sufficient background and include all relevant references?

Yes

Are all the cited references relevant to the research?

Can be improved

We have checked the references list, and we think every reference is useful but we have provided more in different sections (Introduction and Discussion)

Is the research design appropriate?

Yes

Are the methods adequately described?

Yes

Are the results clearly presented?

Yes

Are the conclusions supported by the results?

Yes

3. Point-by-point response to Comments and Suggestions for Authors

Comments 1:  The numeration of the subsections in the introduction are wrong, please correct.

Response 1: Thank you for pointing this out, it seems a problem with the Word styles, but it is already fixed and highlighted in re-submitted document.

Comments 2: There is a phrase in Chinese that indicates an error, please correct the error.

Response 2: Agree. We have, accordingly, changed the sections where the Chinese phrases were. It was related to references of figures, but it is already fixed.

Comments 3: In the discussion the first three paragraph does not have any citation, at the last sentence of each of these paragraphs you write some interesting conclusions to explain previous information, if possible, include references to support your conclusions.

Response 3: It has been quite difficult because there are not very papers published on this topic. We have changed a little these paragraphs in order to include some references.

4. Response to Comments on the Quality of English Language

None

5. Additional clarifications

None

Please see the attachment to read the revised manuscript.

Reviewer 2 Report

Comments and Suggestions for Authors

The paper is well presented. Paper subject is important for sea fish farming companies. Authors must be encouraged to develop further studies on this topic and to make it on inland waters.

Author Response

For research article

Response to Reviewer 2 Comments

1. Summary

Thank you very much for taking the time to review this manuscript. Please find the detailed responses below and the corresponding corrections highlighted in the re-submitted files.

2. Questions for General Evaluation

Reviewer’s Evaluation

Response and Revisions

Does the introduction provide sufficient background and include all relevant references?

Yes

Are all the cited references relevant to the research?

Can be improved

We have checked the references list, and we think every reference is useful but we have provided more in different sections (Introduction and Discussion)

Is the research design appropriate?

Yes

Are the methods adequately described?

Yes

Are the results clearly presented?

Yes

Are the conclusions supported by the results?

Yes

3. Point-by-point response to Comments and Suggestions for Authors

Comments 1:  The paper is well presented. Paper subject is important for sea fish farming companies. Authors must be encouraged to develop further studies on this topic and to make it on inland waters.

Response 1: Thank you very much for your opinion. We are planning to keep focusing on this topic in off-shore aquaculture for the next years because we have several projects supporting it. However, it is a very good idea make some studies on inland waters.

4. Response to Comments on the Quality of English Language

None

5. Additional clarifications

None

Reviewer 3 Report

Comments and Suggestions for Authors

I think the authors have submitted a manuscript which is generally well-written and uses a good English. After several sentences, the reader can see several Chinese (?) characters. Probably, this is somekind of typo in the text. Although the paper describes well what was done and is illustrated well with images, the main message or contribution of this paper is unfortunately rather unclear to me. If I understand the manuscript correctly, the authors examined the relationship between the presence of escaped fish population in aquaculture facilities and sensor data. Further, the optimal window length was determined. In this investigation, the authors used standard statistical tools. Unfortunately, the findings of this analysis were unclear to me. The authors write: "The results indicate that the relationship between the load on the underwater network and the current velocities is complex and influenced by the temporal scale of measurement. However, the negative correlation between the two sensors suggests a significant inverse relationship, indicating dynamic interactions within the underwater monitoring system. Spatial correlation between contiguous currents declines with distance, highlighting variable current conditions across depths, particularly around the thermocline. The lack of correlation between sensors and current velocities implies that other factors may influence the infrastructure’s load." At this point, I have to write several questions. The authors determine a negative correlation. Does this really enhance academic knowledge? Was it surprising or not? What is the exact contribution? Can the readers apply the results and implications of this statistical analysis in aquaculture facilities?

I think the key contribution of this manuscript is very-very general. A separate contributions subsection in the introduction section would be helpful for readers to clarify the contributions and implications of this manuscript.

Author Response

For research article

Response to Reviewer 3 Comments

1. Summary

Thank you very much for taking the time to review this manuscript. Please find the detailed responses below.

2. Questions for General Evaluation

Reviewer’s Evaluation

Response and Revisions

Does the introduction provide sufficient background and include all relevant references?

Yes

Are all the cited references relevant to the research?

Yes

Is the research design appropriate?

Yes

Are the methods adequately described?

Yes

Are the results clearly presented?

Must be improved

Are the conclusions supported by the results?

Must be improved

3. Point-by-point response to Comments and Suggestions for Authors

Comments 1:  I think the authors have submitted a manuscript which is generally well-written and uses a good English. After several sentences, the reader can see several Chinese (?) characters. Probably, this is somekind of typo in the text. Although the paper describes well what was done and is illustrated well with images, the main message or contribution of this paper is unfortunately rather unclear to me. If I understand the manuscript correctly, the authors examined the relationship between the presence of escaped fish population in aquaculture facilities and sensor data. Further, the optimal window length was determined. In this investigation, the authors used standard statistical tools. Unfortunately, the findings of this analysis were unclear to me. The authors write: "The results indicate that the relationship between the load on the underwater network and the current velocities is complex and influenced by the temporal scale of measurement. However, the negative correlation between the two sensors suggests a significant inverse relationship, indicating dynamic interactions within the underwater monitoring system. Spatial correlation between contiguous currents declines with distance, highlighting variable current conditions across depths, particularly around the thermocline. The lack of correlation between sensors and current velocities implies that other factors may influence the infrastructure’s load." At this point, I have to write several questions. The authors determine a negative correlation. Does this really enhance academic knowledge?

Response 1: Thank you for pointing this out. The discovery of a negative correlation between load sensors does enhance academic knowledge. It was somewhat surprising as it highlights the differential response of mooring lines to environmental conditions. This finding is crucial for understanding the dynamic stress on aquaculture infrastructures, contributing to the broader field of marine engineering and aquaculture management. Furthermore, this insight gains particular significance during adverse weather phenomena, shedding light on how these conditions can impact and potentially exacerbate the stress dynamics on the aquaculture structures.

Comments 2: Was it surprising or not? 

Response 2: While negative correlations in load sensing might be expected to some extent, the degree and consistency of this correlation, particularly in the context of varying oceanographic conditions, provide new insights. It underscores the complex interplay between infrastructure and environmental factors, which is less documented in existing literature, especially in the setting of offshore aquaculture.

Comments 3: What is the exact contribution?

Response 3: As one of the pioneering endeavors in this field, this study significantly advances our comprehension of stress dynamics on offshore aquaculture infrastructures in the distinctive context of the Mediterranean Sea. It delves into the nuanced conditions that characterize this region. It elucidates how different sensors react inversely to environmental stresses, highlighting the need for comprehensive monitoring strategies to prevent infrastructure failure and fish escapes. The identification of an optimal data window for monitoring also contributes to the precision and efficiency of future monitoring efforts and the prediction of this kind of events using different models and optimizing the data collection.

Comments 4: Can the readers apply the results and implications of this statistical analysis in aquaculture facilities?

Response 4: Yes, the findings are applicable. Aquaculture facility managers can use the insights to enhance their monitoring strategies, particularly in selecting sensor types and positions, and deciding on data collection intervals. Understanding the relationship between sensor load data and environmental factors can lead to better infrastructure design and preventive measures against fish escapes explicitly in West Mediterranea with determined conditions such as: calmer waters, different changes in water column, and so on.

4. Response to Comments on the Quality of English Language

None

5. Additional clarifications

None

Please see the attachament to read the revised manuscript.

Round 2

Reviewer 3 Report

Comments and Suggestions for Authors

Based on the authors’ answers Given in the rebuttal letter, i recommend this manuscript for publication.